# Development of an ELISA for Measurement of Urinary 3-Hydroxypropyl Mercapturic Acid (3-HPMA), the Marker of Stroke

**DOI:** 10.3390/medsci8030033

**Published:** 2020-08-16

**Authors:** Akihiko Sakamoto, Takeshi Uemura, Yusuke Terui, Madoka Yoshida, Kazumasa Fukuda, Takao Nakamura, Keiko Kashiwagi, Kazuei Igarashi

**Affiliations:** 1Faculty of Pharmacy, Chiba Institute of Science, 15-8 Shiomi-cho, Choshi, Chiba 288-0025, Japan; asakamoto@cis.ac.jp (A.S.); yterui@cis.ac.jp (Y.T.); kkashiwagi@cis.ac.jp (K.K.); 2Amine Pharma Research Institute, Innovation Plaza at Chiba University, 1-8-15 Inohana, Chuo-ku, Chiba 260-0856, Japan; uemura@amine-pharma.com (T.U.); yoshida@amine-pharma.com (M.Y.); 3Graduate School of Pharmaceutical Sciences, Chiba University, 1-8-1 Inohana, Chuo-ku, Chiba 260-8675, Japan; 4Chiba Central Medical Center, 1835-1 Kasori-cho, Wakaba-ku, Chiba 264-0017, Japan; asts-fukuda@ccmc.seikei-kai.or.jp (K.F.); nakamura@ccmc.seikei-kai.or.jp (T.N.)

**Keywords:** acrolein, 3-HPMA, urine, ELISA, stroke, spermine

## Abstract

We previously observed an inverse correlation between stroke and urinary 3-hydroxypropyl mercapturic acid (3-HPMA), an acrolein-glutathione metabolite, through its measurement by liquid chromatography with tandem mass spectrometry (LC-MS/MS). However, the cost of equipment for LC-MS/MS and its maintenance fee is very expensive and a cost-efficient method is required. In this study, we have developed a sensitive enzyme-linked immunosorbent assay (ELISA) system to measure 3-HPMA using a chicken antibody recognizing 3-HPMA-conjugated chicken albumin as antigen. Linearity to measure 3-HPMA was obtained from 0 to 10 μM, indicating that this ELISA system is useful for measurement of urine 3-HPMA. It was confirmed that 3-HPMA in urine of stroke patients decreased significantly compared with that of control subjects using the ELISA system. Using the ELISA kit, it became possible to evaluate the risk of brain stroke by not only plasma but also by urine. These results confirm that shortage of glutathione to detoxify acrolein is one of the major causes of stroke incidence. Our method contributes to maintenance of quality of life (QOL) of the elderly.

## 1. Introduction

Stroke is a serious common pathology and the incidence and severity increase with aging. Therefore, early detection of the risk of stroke and management of the risk factors are crucial to maintain quality of life (QOL), but there were no reliable biomarkers available for the early phase of stroke. We found that increased plasma levels of protein-conjugated acrolein (PC-Acro) and the enzymes responsible for acrolein (CH_2_=CH–CHO) production, polyamine oxidases, were good biomarkers for human stroke [1,2]. It has been reported that silent brain infarction (SBI) increases the risk of subsequent onset of severe brain stroke [3,4]. We also found that measurement of PC-Acro, interleukin-6 (IL-6) and C-reactive protein (CRP) together with age makes it possible to identify SBI, with high sensitivity (84.1%) and specificity (83.5%) [5].

It is thought that cell damage in stroke is caused mainly by reactive oxygen species (ROS) [6,7] such as superoxide anion radical (O_2_•^−^), hydrogen peroxide (H_2_O_2_), and hydroxyl radical (•OH). However, when we compared the toxicity of acrolein and ROS, we found that acrolein was more toxic than ROS [8,9,10]. Furthermore, acrolein was thought of primarily as one of the toxic compounds produced from unsaturated fatty acids by ROS [11], but we found that it was more effectively produced from polyamines, especially from spermine [1,10].

It has been reported that acrolein is conjugated with glutathione and excreted in urine as 3-hydroxypropyl mercapturic acid (3-HPMA) [12]. Thus, we tested whether 3-HPMA in urine of stroke patients decreased compared with that in urine of control subjects, because PC-Acro increased in plasma of stroke patients [1], probably due to the shortage of glutathione, a detoxifying compound of acrolein. We found that 3-HPMA in urine decreased following a stroke, through its measurement by liquid chromatography with tandem mass spectrometry (LC-MS/MS) [13]. However, the cost of equipment for LC-MS/MS and its maintenance fee is very expensive. Thus, we tried to develop a sensitive enzyme-linked immunosorbent assay (ELISA) system to measure 3-HPMA in urine.

## 2. Materials and Methods

### 2.1. Reagents

3-Hydroxypropyl mercapturic acid (3-HPMA) dicyclohexylammonium salt was purchased from Toronto Research Chemicals. Albumin from chicken egg white was purchased from FUJIFILM Wako Chemicals (Osaka, Japan). 1-Ethyl-3-(3-dimethylaminopropyl)carbodiimide (EDC) hydrochloride, Dextran Desalting column, Imject^TM^ Purification Buffer Salts, goat anti-chicken IgY (secondary antibody), and QuantaBlu^TM^ Fluorogenic Peroxidase Substrate Kit were obtained from Thermo Fisher Scientific (Tokyo, Japan). HiTrap IgY Purification HP Column and Can Get Signal^®^ Immunoreaction Enhancer Solution were from GE Healthcare (Tokyo, Japan) and TOYOBO (Osaka, Japan), respectively.

### 2.2. Urine Samples

Urine samples were collected from 51 control subjects without stroke (27 males, 24 females; 58.9 ± 6.5 y, range 40–79 y) and 45 patients with stroke (29 males, 16 females; 71.2 ± 14 y, range 48–92 y). Control subjects were healthy volunteers, living independently at home without apparent history of stroke or dementia. Stroke patients were defined as a sudden onset of nonconvulsive and focal neurological deficit persisting for >24 h and was classified as cerebral infarction or cerebral hemorrhage [14]. Morphological examinations were performed by magnetic resonance imaging (MRI) and computed tomography (CT) on all stroke patients. Urine was collected on day 1 or day 2 after the onset of stroke with procedures approved by the ethics committees of Chiba University and Chiba Central Medical Center. Clinical investigations were conducted in accordance with the Declaration of Helsinki Principles. Informed consent was given in writing by all patients and their caregivers. Urine samples were kept at −80 °C until use.

### 2.3. Imaging

All patients underwent T1- and T2-weighted MRI (magnetic resonance imaging), and some patients underwent fluid attenuated inversion recovery (FLAIR) and CT (computed tomography). All MRI was performed at 5- to 8-mm thickness with 1- to 2-mm slice gap with a 1.5-MRI unit (Signa HiSpeed Infinity, GE Healthcare, Tokyo, Japan). A standard head coil with a receive–transmit birdcage design was used. The maximum size of focal infarcts was measured using 5- or 10-mm length calibration accompanied in each image. NIH stroke scale was evaluated according to the method of Brott et al. [15].

### 2.4. Preparation of Antibody Against 3-HPMA

Antigen was prepared by conjugation of 3-HPMA with albumin from chicken egg white through EDC. Four milligrams of 3-HPMA and 2 mg of albumin from chicken egg white were added in 700 µL of MES [2-(*N*-morphollo)ethanesulfonic acid] buffer (0.1 M MES, 0.9 M NaCl, 0.02% NaN_3_, pH 4.7). Then, 10 mg of EDC was added to the above solution and the mixture was kept for 2 h at room temperature. The conjugation mixture was purified by Dextran Desalting column and Purification Buffer according to the accompanying manual. Chicken monoclonal antibody against 3-HPMA was prepared by MBL (Medical & Biological Laboratories, at Nagano, Japan) using the above antigen (Code Number: CPK-H; LOT: TF0192; Clone Number: 78-1). Conditioned medium of the hybridoma was collected and antibody was purified by HiTrap IgY Purification HP Column according to the manufacturer’s protocol.

### 2.5. Measurement of Polyamines

Urine samples (100 µL) were treated with 100 µL of 10% (*w*/*v*) trichloroacetic acid (TCA), and centrifuged at 15,000*× g* for 10 min. The supernatant (50 µL) was used to measure polyamines, which were measured as described previously [16], using Hitachi high-performance liquid chromatography system on which a TSKgel Polyaminepak column (4.6 × 50 mm) (Tosoh, Tokyo, Japan) was heated to 50 °C. The flow rate of buffer (0.35 M citric acid buffer, pH 5.1, 2 M NaCl and 20% methanol) was 0.42 mL/min. Retention times for putrescine, spermidine and spermine were 6, 12 and 24 min, respectively. Detection of polyamines was by fluorescence intensity with an *o*-phthalaldehyde solution containing 0.06% *o*-phthalaldehyde, 0.4 M boric buffer (pH 10.4), 0.1 M Brij-35, and 37 mM 2-mercaptoethanol. The flow rate of the *o*-phthalaldehyde solution was 0.4 mL/min, and fluorescence was measured at an excitation wavelength of 340 nm and an emission wavelength of 455 nm.

### 2.6. Statistics

Statistical calculations were performed using GraphPad Prism^®^ version 8.1 for Mac (GraphPad Software, La Jolla, CA, USA). Differences between two groups were compared using unpaired *t* test with Welch’s correction. Sensitivity and specificity for stroke patients vs. control subjects were evaluated using a receiver operating characteristic (ROC) curve [17,18].

## 3. Results

### 3.1. Establishment of 3-HPMA Measurement ELISA Kit

Using the purified antibody against 3-HPMA, the level of 3-HPMA in urine was measured by enzyme-linked immunosorbent assay (ELISA). First, 0.05 mL of 40 μg/mL bovine serum albumin was added to each well of a 96-well microplate and kept overnight at 4 °C. Next, 0.05 mL of 3-HPMA solution, pH adjusted to 4.7 by 0.1 M MES, was added to each well in the presence of EDC (1 µmol) and kept for 2 h at room temperature. Then, 100 ng antibody against 3-HPMA/well was added, and finally horse radish peroxidase (HRP) conjugated anti-chicken IgY was added (Figure 1A). The level of anti-chicken IgY interacted with 3-HPMA conjugated albumin was measured by fluorescence intensity (excitation, 325 nm and emission, 420 nm) using QuantaBlu^TM^ Fluorogenic Peroxidase Substrate Kit. As shown in Figure 1B, relative fluorescence unit (RFU) was linear from 0 to 10 μM 3-HPMA.

### 3.2. Decrease in 3-HPMA in Stroke Patients

Using 3-HPMA measurement ELISA kit, 3-HPMA in urine of control subjects and stroke patients was measured using 0.05 mL of urine with pH adjusted to 4.7 by 0.1 M MES. As shown in Figure 2A, the median level of 3-HPMA in 45 stroke patients (0.80 μM) was significantly lower than that in 51 control subjects (2.11 μM). Those values were close to those of previous results, which were measured by LC-MS/MS [13]. The results indicate that the ELISA system is useful for measurement of the urine 3-HPMA level in a cost-efficient way. The ROC curve for the detection of brain stroke is shown in Figure 2B. Area under the curve was 0.7669 (*p* < 0.0001). Sensitivity and specificity were 68.89% and 69.39%, respectively, at the cut off value 1.584. These results confirm that shortage of glutathione is one of the major reasons for stroke incidence.

### 3.3. Decrease in Spermine and Increase in Putrescine and Spermidine in Stroke Patients

We have previously reported that acrolein is produced mainly from spermine (Spm) by spermine oxidase rather than from unsaturated fatty acids by oxidative stress [1,10]. To confirm the idea that acrolein is produced mainly from Spm, polyamine content in urine was also measured. As shown in Figure 3, Spm content was lower and putrescine (Put) and spermidine (Spd) contents were slightly higher in the urine of stroke patients compared to those in control subjects. Consequently, the ratio of Put/Spm and Spd/Spm was significantly higher in the urine of stroke patients. These results confirm the idea that acrolein is produced mainly from spermine.

## 4. Discussion

In this study, we successfully developed a sensitive ELISA system to measure 3-HPMA in urine using antibodies raised against 3-HPMA. We first prepared a polyclonal antibody against 3-HPMA and tried to develop a sensitive ELISA system for the measurement of 3-HPMA in urine. In the system using a polyclonal antibody against 3-HPMA, linearity was obtained in the calibration curve (0 to 10 μM), but sensitivity was low compared with monoclonal antibody. Then, we prepared a monoclonal antibody against 3-HPMA and tried to prepare a calibration curve. The calibration curve obtained for the immunoassay showed a linear range between 0 and 10 µM which is the level of 3-HPMA in urine (Figure 1B) [13]. Using this ELISA kit, the urinary level of 3-HPMA was lower in stroke patients than that in control subjects, similarly to our previous reports obtained through LC-MS/MS measurement [13]. Thus, it became possible to evaluate the risk of brain stroke by not only plasma but also by urine. The area under the ROC curve was 0.7669 which is lower than that of the relative risk value for brain stroke calculated using CRP, IL-6 and protein-conjugated acrolein in plasma (the area under the ROC curve was 0.8940 [19]). A combination of multiple markers can give more effective risk evaluation. If another marker could be found in urine, it will become possible to evaluate the severity of stroke using urine only by combining with 3-HPMA. Urine is easy to collect compared to blood samples, and so with our ELISA system it is possible to provide an effective risk evaluation method for brain stroke that is low in terms of cost and patient’s burden.

We have proposed that acrolein in cells is produced mainly from polyamines, especially from spermine, but not from unsaturated fatty acids [1,9]. We have also shown that the toxicity of acrolein is attenuated by increasing glutathione and decreasing polyamine oxidation [20,21]. In fact, a decrease in Spm and an increase in Put and Spd were observed in the urine of stroke patients compared to control subjects (Figure 3B). A decrease in Spm and an increase in Put due to the upregulation of polyamine degradation pathway was observed in a mouse infarct brain [22]. The urine polyamine contents may reflect tissue polyamine contents and our results suggest that acrolein is produced from polyamines in diseases with cell damage, such as stroke. Since 3-HPMA is a metabolic product of acrolein conjugated with glutathione, our result showing the decrease in 3-HPMA level in patient urine suggests that the degree of detoxification of acrolein by glutathione decreases during the onset of brain stroke.

In the case of plasma, measurement of PC-Acro together with IL-6 and CRP indicates the degree of brain tissue damage [5]. The highest plasma PC-Acro level was observed on the day of stroke onset and decreased in the following days [23]. This observation indicates that acrolein is produced during brain stroke and resulting PC-Acro in plasma is metabolized over several days. In the case of urine, samples were collected 1 or 2 days after the onset of stroke. The significant decrease in 3-HPMA in urine represents the decreased glutathione level during brain infarction since glutathione contributes primarily to the detoxification of acrolein in the brain and is consumed rapidly, possibly on the same day of the stroke onset. Our measurement of 3-HPMA indicates the degree of detoxification of acrolein by glutathione, with the level of 3-HPMA in stroke patients decreasing greatly [13]. For silent brain infarction, it is also expected that the 3-HPMA in urine decreases because of the production of acrolein from small tissue damage and glutathione consumption. Thus, the possibility of finding a silent brain infarction increases significantly by measurement of PC-Acro, IL-6 and CRP in plasma together with 3-HPMA in urine, and contributes to maintenance of quality of life (QOL) of the elderly. Indeed, the measurement of these three biomarkers contributed to the decrease in patients with brain infarction [24]. The number of disease onsets with brain infarction decreased to less than 35% during an evaluation period of 7 years.

## Figures and Tables

**Figure 1 medsci-08-00033-f001:**
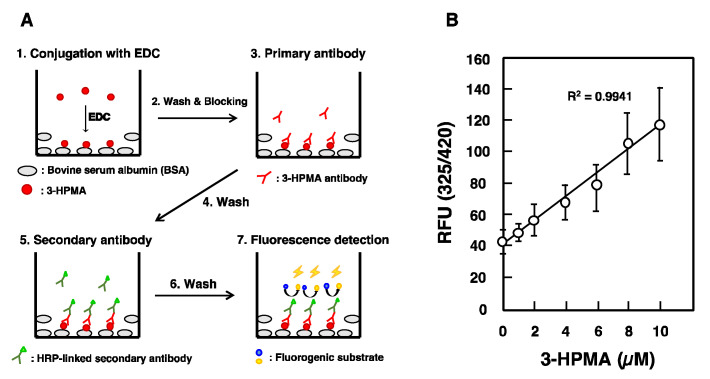
Measurement of 3-hydroxypropyl mercapturic acid (3-HPMA) by enzyme-linked immunosorbent assay (ELISA). (**A**) Schematic presentation of 3-HPMA measurement: 1. Two micrograms of bovine serum albumin was added to each well, and kept overnight at 4 °C. 3-HPMA (0 to 10 μM) in 0.1 mL MES [2-(*N*-morpholino)ethansulfonic acid] buffer (0.1 M MES, 0.9 M NaCl, 0.02% NaN_3_, pH 4.7 containing 1 μmol EDC) was added to each well, and incubated at room temperature for 2 h. 2. After washing with PBS, each well was blocked by 5% skimmed milk in 0.2 mL PBS by incubation at room temperature for 1 h. 3. Antibody against 3-HPMA was purified by HiTrap IgY Purification HP Column according to the manufacturer’s protocol, and 100 ng of purified antibody was added to each well and incubated at room temperature for 1 h. 4. Washing with PBS. 5. Fifty microliters of Amersham ECL anti-chicken IgY, horseradish peroxidase (HRP)-linked species-specific whole antibody was added to each well after 5000-fold dilution with Can Get Signal and incubated at room temperature for 1 h. 6. Washing with PBS. 7. Each well was incubated with QuantaBlu^TM^ Fluorogenic Peroxidase Substrate Kit at room temperature for 15 min. (**B**) Relative fluorescence unit (RFU) was measured at excitation wavelength at 325 nm and emission wavelength at 420 nm.

**Figure 2 medsci-08-00033-f002:**
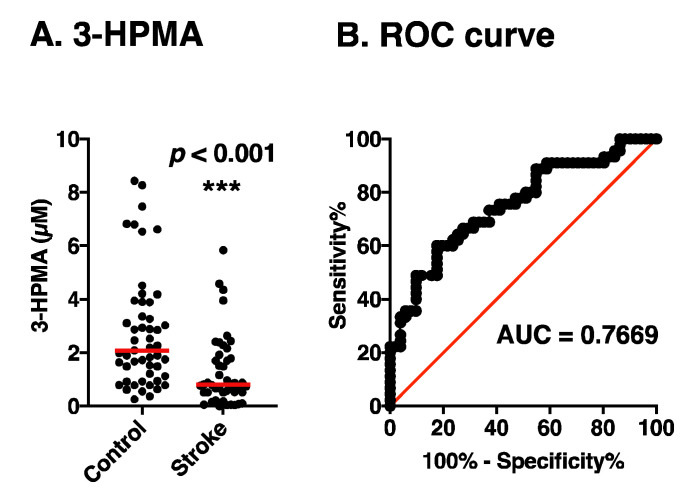
Comparison of the urinary level of 3-HPMA in control subjects and stroke patients. (**A**) The level of 3-HPMA in control and stroke patients was compared. Average value of two experiments is shown with median (horizontal line). The significance of difference was calculated using unpaired *t* test with Welch’s correction. *** *p* < 0.001. (**B**) Receiver operating characteristic (ROC) curve of 3-HPMA for stroke patients vs. control subjects.

**Figure 3 medsci-08-00033-f003:**
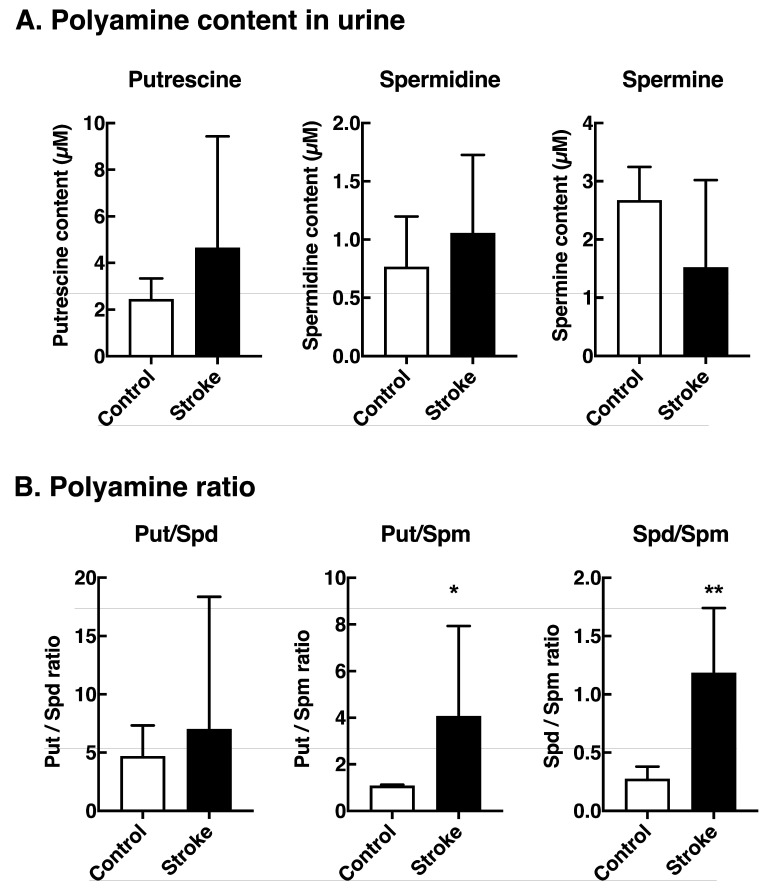
Polyamine content in urine from stroke patients and control subjects. (**A**). Polyamine contents in urine were measured by HPLC [16] and shown as mean + SD. (**B**). The ratio of putrescine/spermidine, putrescine/spermine and spermidine/spermine were calculated and plotted as mean + SD. The significance of difference was calculated using unpaired *t* test with Welch’s correction. Put, putrescine; Spd, spermidine; Spm, spermine. * *p* < 0.05, ** *p* < 0.01.

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
