# Peer review of "Development of an ELISA for Measurement of Urinary 3-Hydroxypropyl Mercapturic Acid (3-HPMA), the Marker of Stroke"

_medsci, 2020, doi:10.3390/medsci8030033_

Round 1

Reviewer 1 Report

Early detection of risk factors is a crucial step in preventing and mitigating the pathological consequence of stroke. Based on the authors’ previous finding that stroke onset is correlated with a reduced unitary 3-hydroxypropyl mercapturic acid (3-HPMA) by LC-MS/MS, this report extended to develop a new ELISA method to meet the detection sensitivity but at a low cost. Overall, the research background is clearly stated, the ELISA method is well designed with the use of chicken monoclonal antibodies against 3-HPMA. The manuscript is well written.

The authors have previously identified three plasma biomarkers to define silent brain infarction (SBI) for early intervention of stroke onset, however, the data points presented in Fig. 2 showed a large overlapping between control and stroke groups, which makes one wonder if ELISA method offers the same sensitivity for the screening of peoples with SBI?

Author Response

#Reviewer 1

Comment 1: The authors have previously identified three plasma biomarkers to define silent brain infarction (SBI) for early intervention of stroke onset, however, the data points presented in Fig. 2 showed a large overlapping between control and stroke groups, which makes one wonder if ELISA method offers the same sensitivity for the screening of peoples with SBI?

Response 1: We would like to thank the valuable comment. We noticed that the overlapping between control and stroke groups which causes lower AUC compared to the combination of three markers in plasma. We expect that combination with other urine marker will provide more efficient biomarker. Discussion was modified accordingly on page 6, lines 200 to 203.

Reviewer 2 Report

The communication by Sakamoto et al. describes the generation and testing of an antibody against the metabolite 3-HPMA, the concentration of which in urine is inversely related to stroke. This ELISA method developed could potentially be used as an economical alternative to the currently employed LC-MS/MS method for measuring the metabolite.  The authors determined that the difference in concentration measured with the antibody in stroke versus healthy patients was in agreement with those previously determined using the instrumental method.

I found the manuscript logically organized and fairly easy to follow.  Background information and description of methods was satisfactory, results well analyzed and I believe that the majority of the conclusions drawn from the results were warranted (see Point number 4 on possible alternate interpretation of spermidine and 3-HMPA levels, below).

Specific comments

  1. Line 176: "...using 3-HPMA-specific antibodies." Was specificity of the antibodies to 3-HPMA actually shown, or should this be rephrased as something like "... using antibodies raised against 3-HPMA."?
  2. Line 181-182: "...linear range between 0 and 10 μM which is the level of 3-HPMA in urine (Figure 1B)." A reference might be good here to indicate the independent study where the data on 3-HPMA level in urine was obtained.
  3. Line 187: "...and so with our ELISA system it is possible to provide a novel risk evaluation method for brain stroke..." Maybe use a word like "effective" in place of "novel" in order to stress the usefulness of the technique.
  4. Line 192: "...a decrease in Spm and an increase in Put were observed in the urine of stroke patients..." 3-HPMA levels actually provide a measure of acrolein, which is derived from spermine (Spm). The fact that the Spm content of urine is lower correlates with the lower 3-HPMA (acrolein) levels in stroke patients, presumably reflecting the fact that there's less Spm to be converted to acrolein. There is no direct data to support this hypothesis. An alternative explanation would be that low Spm levels are caused by a high rate of consumption of Spm in producing some other product, or being degraded. Basically, avoid confusing correlation with causation.

Author Response

#Reviewer 2

Comment 1.        Line 176: "...using 3-HPMA-specific antibodies." Was specificity of the antibodies to 3-HPMA actually shown, or should this be rephrased as something like "... using antibodies raised against 3-HPMA."?

Response 1: We would like to thank valuable comment. We have changed “3-HPMA-specific antibodies” to “antibodies raised against 3-HPMA.”

Comment 2.        Line 181-182: "...linear range between 0 and 10 μM which is the level of 3-HPMA in urine (Figure 1B)." A reference might be good here to indicate the independent study where the data on 3-HPMA level in urine was obtained.

Response 2: A reference was added.

Comment 3.        Line 187: "...and so with our ELISA system it is possible to provide a novel risk evaluation method for brain stroke..." Maybe use a word like "effective" in place of "novel" in order to stress the usefulness of the technique.

Response 3: We have changed “novel” to “effective.”

Comment 4.        Line 192: "...a decrease in Spm and an increase in Put were observed in the urine of stroke patients..." 3-HPMA levels actually provide a measure of acrolein, which is derived from spermine (Spm). The fact that the Spm content of urine is lower correlates with the lower 3-HPMA (acrolein) levels in stroke patients, presumably reflecting the fact that there's less Spm to be converted to acrolein. There is no direct data to support this hypothesis. An alternative explanation would be that low Spm levels are caused by a high rate of consumption of Spm in producing some other product, or being degraded. Basically, avoid confusing correlation with causation.

Response 4: In previous report we have shown that spermine was decreased and putrescine was increased due to the upregulation of polyamine degradation pathway in infarct brain. Moreover, as 3-HPMA is a metabolic product of acrolein conjugated with glutathione, a decrease in urine suggests the degree of detoxification of acrolein decreased during the onset of brain stroke. To avoid confusion, Discussion was modified, on page 6, lines 211-217.